# In Vitro Reduction of Interleukin-8 Response to *Enterococcus faecalis* by *Escherichia coli* Strains Isolated from the Same Polymicrobial Urines

**DOI:** 10.3390/microorganisms9071501

**Published:** 2021-07-14

**Authors:** Gabriella Piatti, Laura De Ferrari, Anna Maria Schito, Anna Maria Riccio, Susanna Penco, Sebastiano Cassia, Marco Bruzzone, Marcello Ceppi

**Affiliations:** 1Department of Surgical Sciences and Integrated Diagnostics (DISC), University of Genoa, 16132 Genova, Italy; amschito@unige.it; 2Department of Internal Medicine, University of Genoa, 16132 Genova, Italy; laura231283@hotmail.it (L.D.F.); anna.maria.riccio@unige.it (A.M.R.); unige14@gmail.com (S.C.); 3Department of Experimental Medicine, University of Genoa, 16132 Genova, Italy; susanna.penco@unige.it; 4Unit of Clinical Epidemiology, Ospedale Policlinico San Martino-IRCCS per l’Oncologia, 16132 Genova, Italy; marco.bruzzone@hsanmartino.it (M.B.); marcello.ceppi@hsanmartino.it (M.C.)

**Keywords:** UTI, urinary tract infection, mixed infection, synergy, CXCL8, interleukin-8, *Escherichia coli*, *Enterococcus faecalis*

## Abstract

Urinary tract infections are often polymicrobial and are mainly due to uropathogenic *Escherichia coli* (UPEC). We previously demonstrated a link among clinical fluoroquinolone susceptible *E. coli* reducing in vitro urothelial interleukin-8 (CXCL8) induced by *E. coli* K-12, polymicrobial cystitis, and pyuria absence. Here, we evaluated whether fifteen clinical fluoroquinolone susceptible UPEC were able to reduce CXCL8 induced by *Enterococcus faecalis* that had been isolated from the same mixed urines, other than CXCL8 induced by *E. coli* K-12. We also evaluated the connection between fluoroquinolone susceptibility and pathogenicity by evaluating the immune modulation of isogenic *gyrA*, a mutant UPEC resistant to ciprofloxacin. Using the 5637 bladder epithelial cell line, we observed that lower CXCL8 induced the most UPEC isolates than K-12 and the corresponding *E. faecalis*. During coinfections of UPEC/K-12 and UPEC/*E. faecalis*, we observed lower CXCL8 than during infections caused by K-12 and *E. faecalis* alone. UPEC strains showed host–pathogen and pathogen–pathogen interaction, which in part explained their persistence in the human urinary tract and coinfections, respectively. Mutant UPEC showed lower modulating activity with respect to the wildtypes, confirming the connection between acquired fluoroquinolone resistance and the decrease of innate microbial properties.

## 1. Introduction

Urinary tract infections (UTIs), the most common bacterial infections in humans, are mainly caused by uropathogenic *Escherichia coli* (UPEC) in both inpatients and outpatients and are often polymicrobial [1,2]. Mixed infections are now recognized as significant burdens for human health and are of interest for the study of overall microbial pathogenicity and for bacteria in particular [3]. The most common group of microorganisms reported in co-infections is that of bacteria, which, differ from those causing the highest global mortality [4].

The pathogenetic mechanisms underlying mixed infections are still poorly defined given their extreme complexity [5]. Different steps and mechanics, even opposing ones, can favour or prevent the onset of co-infections and can be moments and modes for neutral, synergistic, or competitive interactions between different microorganisms.

These different interactions mainly depend on the type of microorganism involved, the different characteristics of the specific strains, and the stages of infection [3,5]. Co-infecting pathogens can interact with each other directly or indirectly while using host resources and overcoming the immune system. They can interact through the availability of quorum sensing receptors (specific to relatives or promiscuous), the availability of nutrients, and through divergent or common nutritional needs [6,7,8].

The great importance of ascending UTIs in humans is due to their high worldwide incidence, notable for both their complicated and uncomplicated forms [1]. The onset and frequency of uncomplicated community-acquired forms, which occur in healthy hosts, are mainly due to the pathogenic characteristics of the bacteria responsible for the infection [9]. Among these, the considerable prevalence of *E. coli*, compared to that of other bacteria that inhabit the human intestine, suggests the paramount importance of the virulence of this species in the genesis of human UTIs [9,10]. Our interest in UTIs is due to the abnormal though habitual presence of microorganisms that already effectively reside in organized niches. We believe that knowledge of how a single microorganism can be present and persist in unusual niches can be improved by understanding how the presence of one microorganism predisposes the host to that of another.

Previously, we found correlations among polymicrobial cystitis and fluoroquinolone susceptible UPEC and their ability to reduce CXCL8 among those cytokines when induced in vitro by the non-pathogen *E. coli*-K12 strain [11]. In the present study, we assessed the interactions between clinical strains of UPEC and *Enterococcus faecalis*, isolated from the same urine, in modulating the CXCL8 response of the 5637 bladder epithelial cell line in vitro. The quinolone resistance determining regions (QRDRs) of *E. coli gyrA* are strongly connected to the scarcity of virulence factors in the species [12,13]. Therefore, we also compared the modulating activity of quinolone-sensitive UPEC strains, which is recognized in the literature as the most virulent [13], with the activity of isogenic mutants in *gyrA*, which are highly resistant to the antibiotic ciprofloxacin.

## 2. Materials and Methods

### 2.1. Patients, Urinary Specimens and Bacterial Strains

Fifteen urine samples representative of urinary infections were selected among those sent to the Clinical Microbiology Laboratory of the San Martino Hospital and had the following characteristics: origin from healthy female outpatients who were under 65 years of age and only suffering from cystitis, the absence of catheter, a lack of pyuria and the presence of ≥10^8^ CFU/mL of a single strain of *E. coli* susceptible to fluoroquinolones, and one sample with a single strain of *E. faecalis*. Urinary specimens were collected following the protocol for routine midstream clean catch collection. Pyuria was considered present with at least 10 leukocytes in 1 mm^3^ of Gram-stained urine, as evaluated by microscopy at magnification × 1000. All *E. coli* strains had a ciprofloxacin minimal inhibitory concentration (MIC) ≤0.06 mg/L. *E. coli* K-12 strain MG1655, and lipopolysaccharide (LPS) mutant was used as a reference stimulating strain [14]. A total of 10 microliters of urine were seeded on Columbia and MacConkey agar plates (Meus) within 1 h of the collection and grown overnight at 37 °C. Bacterial colonies were identified with matrix-assisted laser desorption ionization time-of-flight mass spectrometry (MALDI-TOF) VITEK MS (BioMeriéux Italia S.p.A.). The phenotype of susceptibility to antibiotics was obtained using the automated biochemical testing Vitek 2 (BioMeriéux Italia S.p.A., Grassina, Italy).

### 2.2. Cells Culture

The 5637 bladder epithelial cell line derived from human bladder carcinoma (ATCC HTB-9) used in this study was cultured in RPMI medium (Euroclone) supplemented with 10% fetal bovine serum (Sigma) at 37 °C in a humidified atmosphere of 95% air and 5% CO_2_. Forty-eight hours before stimulation, cells were seeded into 24-well plates at 1.5 × 10^5^ per well in 1 mL of antibiotic-free cell culture medium and allowed to grow to confluence over two days. On the day of the stimulation, the confluent cell monolayers were washed with sterile phosphate-buffered saline and 1 mL of fresh medium was applied.

### 2.3. Interleukin-8 Stimulation Assay

Four types of monomicrobial infections (containing the UPEC wildtype and mutant strains, *E. coli* K-12, and *E. faecalis*) and four types of mixed infections (containing the UPEC wildtype and mutant strains plus the common *E. coli* K-12, the UPEC wildtype, and mutant strains plus *E. faecalis*) were performed as previously described [15]. First, the bacteria grown overnight in Luria broth cultures were diluted to 1 × 10^8^ CFU/mL. Ten microliters of each bacterial suspension (in the coinfection experiments) and twenty microliters (in the mono-infection experiments) were placed in single wells containing 5637 epithelial cells to obtain a concentration of 1 × 10^6^ CFU/mL. We assessed the production of CXCL8 both by uninfected cells (to evaluate their baseline production) and by infected cells after 4 h of infection. Bacterial concentration and incubation time were chosen to achieve the best bacterial stimulation relative to the baseline bladder cell interleukin production with viability >95%, which was verified by trypan blue staining (data not shown).

### 2.4. ELISA

The quantities of CXCL8 in the collected supernatants were determined with the ELISA determination method (Quantikine ELISA-bio-techne). The results of the stimulation (pg/mL) were obtained as the average of two experiments performed for each monomicrobial and mixed infection. We reported the differences obtained between the different monomicrobial infections. We reported the reductions obtained by the mixed infections, expressed as percentages, with respect to the monomicrobial ones. We arbitrarily did not consider values <5 percent as differences.

### 2.5. Generation of UPEC Mutants

Overall, Mutagenesis experiments were performed as already described [16,17] but with the following modifications: First, for each of the seven strains, one colony was grown at 37 °C for 24 h in Luria broth without ciprofloxacin and subsequent seeding was repeated five times. For assaying the mutations, approximately 10^8^ cells corresponding to 150 microliters from each culture were plated in duplicate on Luria agar containing ciprofloxacin at increasing serial concentrations by multiples of two, i.e., from 0.064 mg/L to 32 mg/L. Every 24 h, the colonies that had grown on any antibiotic containing plates were seeded onto a plate with the next higher antibiotic concentration. For each strain, we performed experiments at each antibiotic concentration, obtaining the same results for all. We aimed to obtained mutant UPEC strains resulting from post-exposure mutations and considered only those mutations that conferred resistance. Consequently, we did not perform reconstruction assays or mutation rates. We did not exclude cultures with filaments that were >5% of the total cells generated from seeding, and that increased mutant frequency. We collected all of the strains that achieved the ciprofloxacin resistance phenotype. For the stimulation assay and DNA sequencing, we utilized the colonies grown at the highest concentration of ciprofloxacin.

### 2.6. MIC Determination to Ciprofloxacin of UPEC Strains

The determination of the minimal inhibitory concentrations (MICs) against ciprofloxacin was performed with the twofold serial dilution method. MIC values were calculated to verify the automated data of the UPEC wildtype strains and the resistance phenotype of the seven isolates grown at the different concentrations of ciprofloxacin. *E. coli* ATCC 25922 strain was used as the control strain.

### 2.7. Biotypes and Virulence Genotyping/Phenotyping of UPEC Strains

UPEC isolates were grouped into A, B1, B2, and D phylogenetic biotypes by using multiplex PCR as previously described [18]. The presence of the virulence genes *cnf1*, *hlyA*, *fimA*, and *papC* encoding cytotoxic necrotizing factor-1, hemolysin, type 1 fimbriae, and P fimbriae, respectively, was investigated by means of PCR as described elsewhere [19,20]. Hemolisin production was visually assessed by the appearance of a clear halo on 5% sheep blood agar plates after overnight culture at 37 °C. Type 1 fimbriae were investigated using the mannose sensitive agglutination of *Saccharomyces cerevisiae*, and P fimbriae were shown through P blood group-dependent hemagglutination.

### 2.8. Sequencing of gyrA QRDR of UPEC Strains

The DNA of the seven UPEC strains treated with a low dose of ciprofloxacin in vitro and grown at the highest concentration of ciprofloxacin was sequenced to check for mutations in *gyr*A QRDR. The strains, numbered from 9 to 15, were cultured in Luria broth and seeded on Luria agar plates. One colony per strain was picked, dispersed in water, and boiled to extract the DNA. After centrifugation, the supernatant was amplified using the following primers: *gyrA*-F: 5′-CCA TGA ACG TAC TAG GCA ATG A-3′, base from ATG 152-173; *gyrA*-R: 5′-AAT TTT CGC CAG ACG GAT T-3′, base from ATG 390-372. (TIB-Molbiol, Genoa Italy) DNA topoisomerase (ATP-hydrolyzing) subunit A NC_010473—REGION: 2428430 2425803. Amplicons were sequenced using the Sanger method (LGC Genomics GmbH, Berlin, Germany). We used the *gyrA* sequence of wildtype UPEC strains and *E. coli* K-12 as references. We did not consider mutations that appeared to be coming from a sequencing error since they occurred in each reference genome. We also removed mutations that were too close to one another (less than 51 bp apart).

### 2.9. Statistical Analysis

The nonparametric Spearman correlation coefficient was computed to verify the degree of association among the four comparisons (indices) between results from different infections and coinfections [21]. Since the experimental design included repeated tests on the same subject, the Wilcoxon matched-pairs signed-ranks test was applied for all of the comparisons between groups of infections [22]. Differences between the seven strains among the indices before and after induction were compared using the Kruskal–Wallis equality-of-populations rank test [23]. To adjust for multiple comparisons, Bonferroni correction was applied, i.e., a critical *p*-value estimated as a ratio between the type I error, usually 0.05, and the number of comparisons that have been carried out. Only *p*-values lower than the critical value were considered statistically significant. STATA software was used for all statistical analyses (StataCorp. 2015. Stata Statistical Software: Release 14. College Station, TX, USA: StataCorp LP). A box-plots graphical model was used for the visual inspection of the distribution of data and of any possible outliers [24].

## 3. Results

### 3.1. Clinical UPEC Induce Less CXCL8 Than E. coli K-12 during In Vitro Infections

We first aimed to discover any differences between fifteen clinical UPEC strains and the common non-pathogenic *E. coli* K-12 in host–pathogen interaction. We compared CXCL8 production of epithelial cells infected with each UPEC strain with that of cells infected with K-12. Of the analyzed strains, thirteen induced lower CXCL8 production than K-12 (mean difference 57.5%, range 31–90%), while two strains stimulated a higher CXCL8 release than K-12 (Figure 1A).

### 3.2. Clinical UPEC Limit CXCL8 Induced by E. coli K-12 during In Vitro Coinfections

After observing that clinical UPEC had a lower stimulatory effect than the non-pathogenic *E. coli* K-12, we evaluated whether the same strains were also able to modify the cellular response to K-12 during coinfections. We compared CXCL8 produced by cells co-infected with each UPEC plus the common *E. coli* K-12 to CXCL8 produced by cells that were only infected with *E. coli* K-12. Twelve coinfections induced less CXCL8 than infection with *E. coli* K-12 alone (mean reduction 45.7%, range 26–64%). One coinfection induced the same quantity of CXCL8 induced by *E. coli* K-12, and two coinfections induced a greater quantity of CXCL8 (Figure 1A).

### 3.3. Clinical UPEC Induce Less CXCL8 Than Clinical E. faecalis during In Vitro Infections

Once the eligibility of the UPEC strains to contribute to polymicrobial infections was confirmed to be possible, we evaluated the differences between the UPEC and *E. faecalis* strains on in vitro host–pathogen interaction. We aimed to learn which of the microorganisms present in the same urine had behavior similar to the non-pathogenic K-12 strain and which was different. We compared the CXCL8 production of epithelial cells infected with each UPEC strain to the CXCL8 produced by the cells that had been infected with the corresponding *E. faecalis*. Eleven UPEC strains induced less CXCL8 than the corresponding *E. faecalis* strain (mean difference 41.3%, range 16–82%). One UPEC strain induced the same quantity of CXCL8, and three UPEC strains induced more than the corresponding *E. faecalis* strain (Figure 1B). Most UPEC strains showed behaviour that was very different from K-12, while most *E. faecalis* strains showed activity that was very similar to K-12.

### 3.4. Clinical UPEC Limit CXCL8 Induced by Clinical E. faecalis during In Vitro Coinfections

We aimed to determine if the UPEC strains that induced lower CXCL8 production than *E. coli* K-12 and the corresponding *E. faecalis* and that also limited the production of K-12 were also able to reduce the CXCL8 production caused by *E. faecalis* strains. We compared the amounts of CXCL8 produced by cells that been coinfected with UPEC and *E. faecalis* that been isolated from the same urine to those produced by cells infected with *E. faecalis* alone. Twelve coinfections induced less CXCL8 than monomicrobial infections (mean reduction 39.3%, range 7–64%). Three coinfections induced more CXCL8 than *E. faecalis* alone (Figure 1B).

The Spearman coefficient describes the association between the four global indices (all 15 UPEC strains) (Table 1).

The index for UPEC/*E. faecalis* shows a strongly positive correlation with UPEC plus *E. faecalis*/*E. faecalis*. The weakest correlation is between UPEC/K-12 and UPEC plus K-12/K-12. The remaining coefficients show low correlations. The inferential evaluation of the matched-pairs test shows that the highest median belongs to the UPEC/*E. faecalis* group while the lowest one belongs to the UPEC/K-12 group (Table 2).

Since no value is <0.008 (critical *p*-value), it is possible to state that there are no significant differences between the indices obtained from the different stimulations. The graphic representation in the box-plots shows that, out of fifteen strains, the outliers are very few, approximately one to three for each index. This confirms the overall uniformity of the results between the different UPEC and *E. faecalis* strains and the different stimulations (Figure 2).

It is interesting to note that outlier strains are mostly the same (numbers 5, 6, in UPEC/K-12; 6, 7 in UPEC plus K-12/K-12; 13 in UPEC/*E. faecalis*; 5 in UPEC plus *E. faecalis*/*E. faecalis*) and the strains that exceed the 75 percentile in one index mostly exceed it in the other ones.

### 3.5. Effect of Treatment with Ciprofloxacin on Ciprofloxacin Susceptibility and Virulence Factors of UPEC

Seven UPEC became ciprofloxacin resistant after treatment with ciprofloxacin (MIC ranging from 8 mg/L to 32 mg/L). In particular, the MICs were 8 mg/L for two strains, 16 for three, and 32 for two. Two strains lost the phenotypic expression of type 1 fimbriae, i.e., the mannose sensitive agglutination. Two strains were biotype B2, three strains were biotype B1, one strain was biotype A, and one strain was biotype D. The comparison between the VFs and biotypes of resistant UPEC and their mutational attitude did not lead to any significant findings.

### 3.6. Mutations in the QRDR of gyrA of Seven UPEC Induced to FQ Resistance

We sequenced the genome’s QRDR fragment for resistant strains to ensure that the resistance phenotype was linked to the presence of mutations. The nucleotide sequences of all seven resistant UPEC showed at least one nonsynonymous single nucleotide polymorphism (SNP) mutation in the QRDR region of *gyrA* at Ser83 or Asp87 of the fragment analyzed.

All SNPs at Ser83, which was present in five isolates, gave Leu (Leucine); two SNPs at Asp87, which was present in two isolates, gave Gly (Glycine); one SNP at Asp87, which was present in one isolate, gave His (Histidine); and another one, which was present in one isolate, gave Asn (Asparagine). Hence, two out of the four SNPs were shared by different isolates, while two SNPs were specific to one. Two isolates had at least double mutations of gyrA (having not been sequenced up to codon 106) and MIC 32 mg/L. Among strains with one single mutation, three had MIC 16 mg/L and two had MIC 8 mg/L. All of the UPEC isolates showed wildtype sequences of *gyrA* before induction.

### 3.7. UPEC Mutants Induce More CXCL8 Than Wildtypes during In Vitro Infections

We assessed whether *gyrA* mutant ciprofloxacin resistant UPEC showed differences in vitro host–pathogen interaction compared to the wildtype strains. We compared the CXCL8 produced by cells that had been infected with wildtype UPEC to the CXCL8 produced by cells infected that had been with mutants. Six mutants induced more CXCL8 than the corresponding wildtype (average difference 22%, range 17–27%) (Figure 3).

The difference in both six out of seven UPEC and *E. coli* K-12 stimulation between the CXCL8 values decreased from the average difference of 61% to 51.4% (strains 9–15) when we utilized the mutant strains. The difference between CXCL8 values in one UPEC and *E. faecalis* did not change, disappeared in a different UPEC, decreased in two UPEC and *E. faecalis*, and the values another two UPEC were even higher than by *E. faecalis* using the mutant strains. We did not consider mutant number 13 because the wildtype strain, though it was already slightly more stimulating than the *E. faecalis* strain.

### 3.8. UPEC Mutants Modulate Less Than Wildtypes during In Vitro Coinfections

Finally, we compared the ability of wildtype UPEC with the ability of mutants in modulating CXCL8 production induced by the common K-12 and by each *E. faecalis* isolated in coinfections of the epithelial cells. Five K-12 coinfections performed using mutants reduced CXCL8 induced by K-12 less than those performed with wildtype analogues, with a mean reduction of 41.4% (strains 9–15) to 33.4%. The remaining coinfections abolished the difference with K-12 alone (Figure 4A). *E. faecalis* coinfections, performed using two mutant UPEC modulated stimulation with *E. faecalis* had less difference than wildtypes coinfections did. Two coinfections performed with mutants were not successful in the modulation activity. The mutant counterparts of two UPEC strains, which in coinfections induced more CXCL8 than *E. faecalis* alone, gave even higher values in mixed stimulations (Figure 4B).

The results of the in vitro stimulation obtained from seven wildtype UPEC and compared with their mutant counterparts showed low *p*-values for each sign test, which suggests that the mutation has the effect of reducing the difference in each group. However, by applying the Bonferroni correction, only the overall comparison acquires statistical significance (Table 3).

When performed individually for each strain, the same evaluation showed that the strongest modulatory effect of the mutation occurred in strain number 13 (median difference 37.9%) while the weakest effect occurred in strain 10 (median difference 2.2%). Overall, the Kruskal–Wallis test did not reveal significant heterogeneity among the strains (*p* = 0.061) (data not shown). The graphic representation in the box-plots shows that out of seven clinical strains, the outliers are very few (wildtype and mutant 13 in UPEC/*E. faecalis*; mutant 15 in UPEC plus K-12/K-12). This confirms the overall uniformity of the results of the tests, between both the UPEC strains and the different types of stimulation (Table 3, Figure 5).

## 4. Discussion

In the field of polymicrobial infections affecting the urinary tract in humans, many works have focused on the mechanisms underlying bacterial interactions. As far as we know, the present work is the first that describes *E. coli* as the organism responsible for a mixed infection rather than as one that is facilitated by other bacteria, such as *E. faecalis* and *Proteus mirabilis* [6,25,26,27]. The frequency of *E. coli* and of its polymicrobial forms is substantial in both uncomplicated and complicated UTIs [1,2]. This led us to investigate how *E. coli* can affect not habitual mucosa and whether it could be responsible for the synergy favoring mixed infections.

In this study, we showed that most of the clinical UPEC we analyzed induced less CXCL8 than the *E. faecalis* strains isolated from the same mixed urine in vitro and, during coinfection, reduced the CXCL8 induced by *E. faecalis* itself. These results, which were also obtained when *E. coli* K-12 was used in place of *E. faecalis*, lead us to attribute the main role in subverting the early immune response to *E. coli*, which we stimulated here in vitro. Among the pattern recognition receptors (PRRs), different Toll-like receptors (TLRs) possess several common adaptor molecules, such as the MyD88 [28]. Thus, TLR4 and TLR2, while recognizing different microbe-associated molecular patterns (MAMPs), endotoxins, and Gram-positive bacterial lipoproteins (BLP), can also act in concert as agonists [29]. Considering that BLP tolerance generates endotoxin tolerance through a defect in MyD88/IRAK in mice, it is likely that inhibition and stimulation act through the same pathway [30,31,32], explaining the possibility of cross-suppressive bacterial activity in the bladder epithelial cells, where both stimulating and tolerance activities occur [33]. We suppose that by inducing anergy in the urothelium, a bacterial species could confer a local advantage to initiate colonization in another microorganism that has not yet been exposed. In this context, *E. faecalis* appears to behave similarly to K-12, a non-pathogenic organism, which may benefit from the virulence of another non-pathogenic organism. We think that the differences between our results and those from previous studies are likely due to the multiple variables that, in turn, depend on the multifactorial nature of polymicrobial infections. In the work showing that *E. faecalis* prevents NF-kB signaling, a bacterial interaction from which *E. coli* benefits, from occurring in macrophages of the bladder submucosa, where bacterial presence causes inflammation and worsening of the disease [26]. In fact, the recruitment of neutrophils, which may hinder the bacterial presence in the lumen, can instead favor damage progression in the submucosa [27,34]. In order to avoid variables linked to successive phases, we focused on the site and time when the interaction between the MAMPs and pattern recognition receptors (PRRs) occurs for the first time, acting as very early innate defenses [34]. Moreover, in the above-mentioned work, immune modulation was investigated after the placement of a urinary catheter in animals, which can mask the very first stage of host–pathogen interaction. Urinary catheterization, causing biofilm and inflammatory response in addition to those elicited, for instance, by *E. faecalis*, can cover the natural bacterial capability to colonize sites different from its own habitat, which is our greatest interest [35,36,37]. For these reasons, we excluded urines collected from catheters from our study and, to further reduce the weight of variables, we investigated *E. coli* and *E. faecalis* isolated with similar clinical evidence, i.e., where interaction could have taken place. We collected urine where both species were simultaneously present and where leukocytes were missing from healthy female outpatients.

Our results showed a sort of homogeneity within both the *E. coli* and *E. faecalis* groups. The strains endowed with modulatory features were in each analysis Infections and coinfections performed with the clinical strains UPEC and *E. faecalis* isolated from the same urine without K-12, showed an especially strong association between them (0.786). The few strains falling outside this mode proved consistent in the different assessments. In the work of Wang, et al., only a minor portion of the *E. faecalis* strains that were evaluated were able to down-regulate CXCL8 secretion, implying a strain dependent immunomodulation in the host, which may in part explain the different results among different works [38].

The connection between the acquisition of antibacterial resistance and the decrease of virulence as a fitness cost is ascertained in *E. coli* and in other bacterial genera [12,13,39,40]. Since studies based on the reversed features of isogenic strains are few, it remains unclear whether the lower pathogenicity of resistant strains, compared to susceptible bacteria, is attributable to different growth or different patterns of interaction with the innate immune system [41]. We obtained ciprofloxacin resistant isogenic UPEC (with MIC at least 8 mg/mL) that developed mutations corresponding to those reported worldwide for spontaneous and in vitro induced resistant *E. coli*. In most mutant strains, we observed SNP Ser83Leu, the most frequent documented amino acid substitution in *gyrA* [13,42]. By comparing wildtype with mutant strains, we observed a negative impact in the mutations on their ability to escape and modulate the innate proinflammatory responses.

The lack of correlation between the UPEC strains modulatory activity and virulence factors *sensu stricto*, as previously demonstrated [11], is consistent with other investigations. The type 1 piliated *E. coli* K-12, a mutant of the MAMP LPS, loses modulatory activity, while NU14-1, an isogenic mutant of NU14 for *fimH*, continues to possess that modulatory activity [15,33]. Moreover, insertional mutations in the *rfa* and *rfb* operons and in the *surA* gene are known to destroy LPS and outer membrane protein biosynthesis, both by communicating with the host’s PPRs, abolishing the ability of UTI89 to suppress cytokine induction [15]. These preconditions also indicate the limitations of our study. First, we did not sequence the genes directly involved in the synthesis of LPS or membrane proteins, nor those encoding inhibitory homologs of Toll/interleukin-1 receptor domains, which subvert Toll-like receptor signaling in UPEC strains, *Brucella melitensis* and *E. faecalis* [43,44]. Second, the in vitro assessment of the growth pattern of the *gyrA* mutant UPEC versus its wildtype counterpart could be performed to evaluate if the decrease of modulation by mutant resistant strains is due to their lower survival when compared to the susceptible wildtypes. Therefore, additional work is required to search for mutations in the mentioned genes and to evaluate whether the changes in bacterial features are indirectly due to the affected replicative metabolism of *gyrA* mutant *E. coli*.

## 5. Conclusions

UPEC strains isolated from human polymicrobial urine can be responsible for their own presence in the urinary tract and for infection and coinfection through host–pathogen and pathogen–pathogen interactions. These interactions, which were reduced after *gyrA* mutations, seem to be part of the full capabilities of *E. coli*.

## Figures and Tables

**Figure 1 microorganisms-09-01501-f001:**
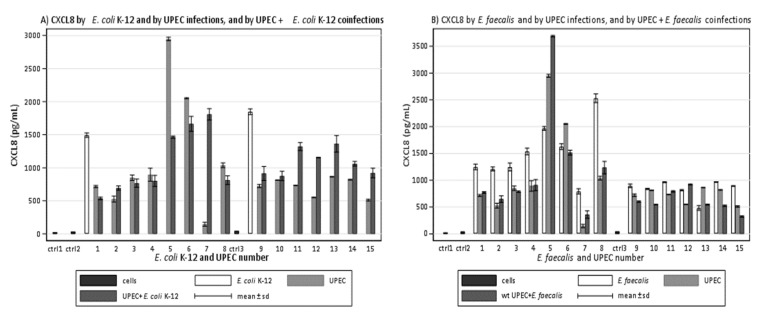
Amounts (pg/mL) of CXCL8 induced in 5637 bladder epithelial cell line by fifteen UPEC: (**A**) CXCL8 induced by UPEC strains and by UPEC strains plus the common *E. coli* K-12 compared to CXCL8 induced by the common *E. coli* K-12 alone; (**B**) CXCL8 induced by UPEC strains and by UPEC strains plus *E. faecalis* isolated from the same urine compared to CXCL8 induced by *E. faecalis* alone.

**Figure 2 microorganisms-09-01501-f002:**
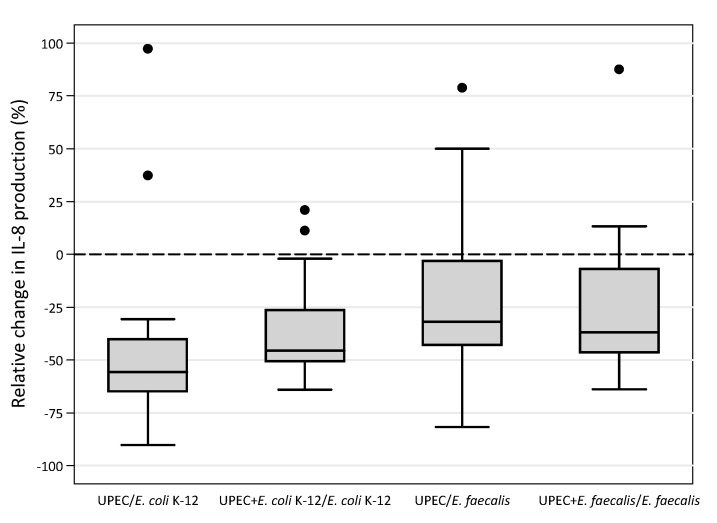
The four indices between values of CXCL8 induced in the 5637 bladder epithelial cell line by fifteen UPEC during in vitro monomicrobial and mixed infections (see Figure 1) displayed as box-plots. The lines inside boxes represent the median value of the observations, and the lower and upper sides of the boxes are the 25th and 75th percentile, respectively. Circles beyond the last bar represent the outliers as they exceed the 75th percentile plus one and half times the interquartile range.

**Figure 3 microorganisms-09-01501-f003:**
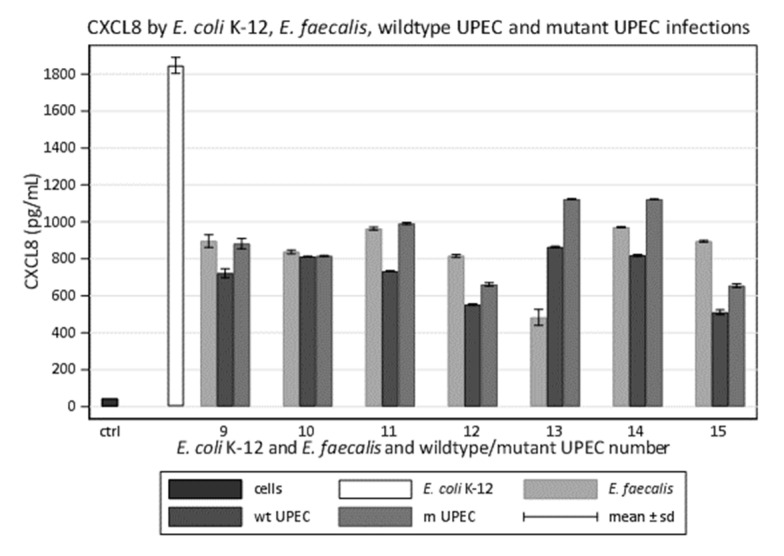
Values of CXCL8 induced in 5637 bladder epithelial cell line by monomicrobial infections from the seven wildtype UPEC compared to CXCL8 induced by monomicrobial infections from the seven mutant analogues, with the common *E. coli* K-12 and with *E. faecalis* strains isolated from the same urine. wt, wildtype; m, mutant.

**Figure 4 microorganisms-09-01501-f004:**
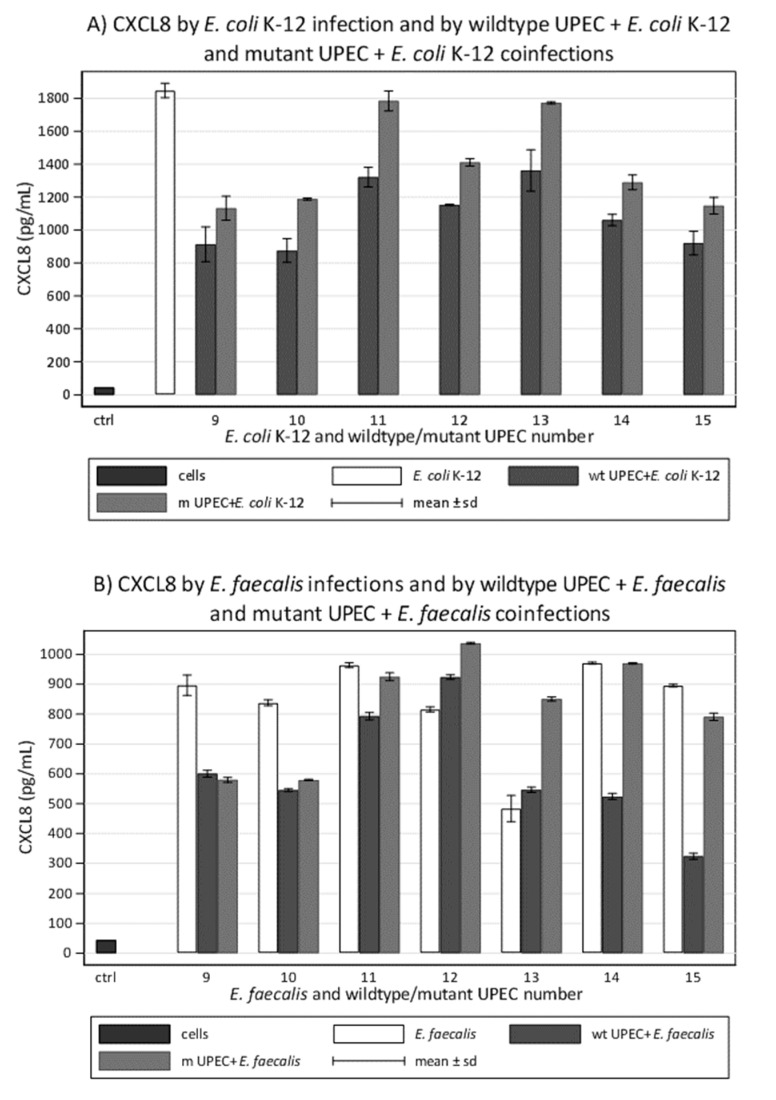
Values of CXCL8 induced in 5637 bladder epithelial cell line by polymicrobial infections with the seven wildtype UPEC and the seven mutant analogues: (**A**) CXCL8 induced by wildtype and mutant UPEC plus the common *E. coli* K-12 compared to CXCL8 induced by *E. coli* K-12 alone; (**B**) CXCL8 induced by wildtype and mutant UPEC plus *E. faecalis* isolated from the same urine compared to CXCL8 induced by *E. faecalis* alone. wt, wildtype; m, mutant.

**Figure 5 microorganisms-09-01501-f005:**
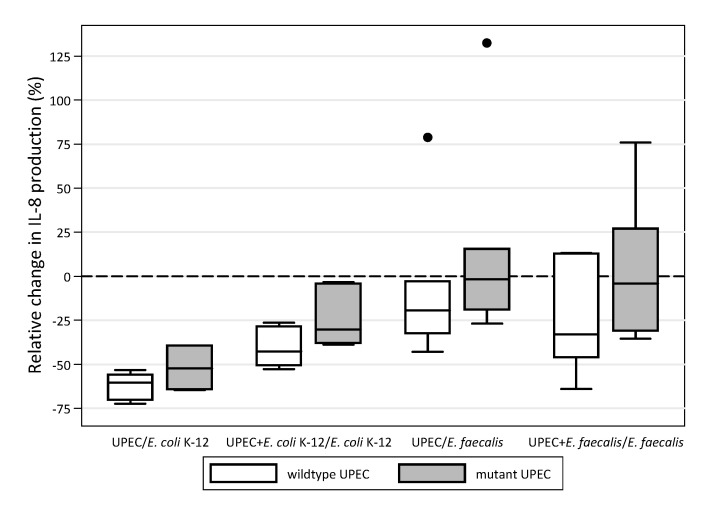
The four indices between the values of CXCL8 induced in the 5637 bladder epithelial cell line during in vitro monomicrobial and mixed infections (see Figure 1) are displayed as differently coloured box-plots for the seven wildtype and the seven isogenic mutant UPEC. The lines inside boxes represent the median value of the observations, and the lower and upper sides of the boxes are 25th and 75th percentile, respectively. The circles beyond the last bar represent the outliers as they exceed the 75th percentile plus one and a half times the interquartile range.

**Table 1 microorganisms-09-01501-t001:** Spearman correlation coefficient for the overall six pairwise associations among indices obtained from the different infections of the 5637 bladder cell line.

Infection	UPEC/ K-12	UPEC+K-12/ K-12	UPEC/ *E*. *faecalis*	UPEC+E. *faecalis*/ *E*. *faecalis*
*^a^* UPEC/K-12	1.000			
*^b^* UPEC+K-12/K-12	0.161	1.000		
*^c^* UPEC/*E*. *faecalis*	0.432	0.286	1.000	
*^d^* UPEC+*E. faecalis*/*E. faecalis*	0.357	0.343	0.786 *	1.000

*^a^* index between CXCL8 induced by 15 UPEC and by *E. coli* K-12; *^b^* index between CXCL8 induced by 15 UPEC plus *E. coli* K-12 and by *E. coli* K-12; *^c^* index between CXCL8 induced by 15 UPEC and by 15 *E. faecalis*; *^d^* index between CXCL8 induced by 15 UPEC plus 15 *E. faecalis* and by 15 *E. faecalis*; * statistically significant according to the Bonferroni method, critical *p*-value = 0.05.

**Table 2 microorganisms-09-01501-t002:** Inferential evaluation of indices and related median, 25th, and 75th percentiles for the overall six pairwise associations among indices obtained from the different infections of the 5637 bladder cell line.

	Median (%)	Percentiles 25/75 (%)			Median (%)	Percentiles 25/75 (%)	Observed * *p*-Value
*^a^* UPEC/K-12	−55.7	−64.8/−40.2	vs.	UPEC+K-12/K-12	−45.6	−50.6/−26.3	0.334
				UPEC/*E. faecalis*	−31.8	−42.9/−3.0	0.054
				UPEC+*E*. *faecalis*/*E. faecalis*	−36.9	−46.4/−6.7	0.061
*^b^* UPEC+K-12/K-12	−45.6	−50.6/−26.3	vs.	UPEC/*E. faecalis*	−31.8	−42.9/−3.0	0.027
				UPEC+*E*. *faecalis*/*E*. *faecalis*	−36.9	−46.4/−6.7	0.140
*^c^* UPEC/*E. faecalis*	−31.8	−42.9/−3.0	vs.	*^d^* UPEC+*E*. *faecalis*/*E*. *faecalis*	−36.9	−46.4/−6.7	0.733

*^a^* index between CXCL8 induced by 15 UPEC and by *E. coli* K-12; *^b^* index between CXCL8 induced by 15 UPEC plus *E. coli* K-12 and by *E. coli* K-12; *^c^* index between CXCL8 induced by 15 UPEC and by 15 *E. faecalis*; *^d^* index between CXCL8 induced by 15 UPEC plus 15 *E. faecalis* and by 15 *E. faecalis*; * critical *p*-value = 0.008 estimated using the Bonferroni method.

**Table 3 microorganisms-09-01501-t003:** Inferential evaluation of indices and related median, 25th, and 75th percentiles for the overall associations through the comparison of seven wildtype and mutant UPEC.

	Wildtype UPEC		Mutant UPEC		
	Median (%)	Percentiles 25/75 (%)	Median (%)	Percentiles 25/75 (%)	Observed *p*-Value
*^a^* UPEC/K-12	−60.3	−70.1/−55.7	−52.2	−64.2/−39.2	0.018
*^b^* UPEC+K-12/K-12	−42.6	−50.6/−28.5	−30.2	−37.8/−4.0	0.018
*^c^* UPEC/*E*. *faecalis*	−19.4	−32.3/−3.0	−1.6	−18.9/15.6	0.018
*^d^* UPEC+*E*. *faecalis*/*E*. *faecalis*	−33.0	−46.0/13.0	−4.0	−30.9/27.1	0.028
Overall	−40.1	−55.0/−20.5	−25.2	−39.1/−1.9	<0.001 *

*^a^* index between CXCL8 induced by 7 UPEC and CXCL8 induced by *E. coli* K-12; *^b^* index between CXCL8 induced by 7 UPEC plus *E. coli* K-12 and by *E. coli* K-12; *^c^* index between CXCL8 induced by 7 UPEC and CXCL8 induced by 7 *E. faecalis*; *^d^* index between CXCL8 induced by 7 UPEC plus 7 *E. faecalis;* and CXCL8 induced by7 *E. faecalis*; * statistically significant according to the Bonferroni method, critical *p*-value = 0.010.

## Data Availability

The data presented in this study are available upon request from the corresponding author.

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
