# Peer review of "In Vitro Reduction of Interleukin-8 Response to Enterococcus faecalis by Escherichia coli Strains Isolated from the Same Polymicrobial Urines"

_microorganisms, 2021, doi:10.3390/microorganisms9071501_

Round 1
Reviewer 1 Report
The article reports the propensity of various uropathogenic bacteria recovered from polymicrobial infected urine samples, when applied alone or in combination, to stimulate a bladder epithelial cell line to secrete IL-8. The supposition was that some bacteria could inhibit the stimulation of the IL-8 production by other bacteria. The samples were collected from patients without pyuria, so there ought to be a good chance that bacteria either fail to stimulate IL-8 or inhibit IL-8 synthesis or secretion.
I have concerns over the manner in which the data is presented for, 2 reasons, 1.the same data is used in multiple panels, giving the impression that the experiment was repeated when it was probably not. Therefore, multiple panels could be condensed into a single figure. For example, Figure 1, it is unnecessary to repeat the E. coli K-12 bar with every sample because the data is derived from the same experiment. The “ctrl” is only shown once, so could the E. coli K-12. The E. coli K-12 bars in Fig 1A and B are the same data, so combine the UPEC+E.coli K-12 data in panel B into panel A. The same data for the E. coli in panel A is presented as if it is a different experiment when compared to the UPEC+E.coli K-12 in panel B. Then again, the UPEC data in panel A is used in panel C. (And the data is used again in Figure 4.) Unless they are all included in the same figure, the impression is that data is being mixed between multiple experiments. This latter possibility is concerning because the authors actually show there is considerable variability between experiments when in Figure 1A the E. coli K-12 is done twice, once for the comparison with samples 1-8 and a second time for comparing with samples 9 through 15. The difference in the IL-8 levels between these two experiments is likely statistically significantly different (1500 compared to 1800, tiny standard deviations). So either all the bars appear in one figure to show the experiment was done with all the bacteria at the same time, or repeat each bacteria in each panel, because the variation between experiments for the same bacteria is too great to just share the data from an earlier experiment.
I am also concerned that the differences, in many case rather small despite being statically significant, are likely not biologically relevant. Measure the cell supernatants in a “functional” assay, directly measuring neutrophil migration. Migration occurs to IL-8 concentrations spanning almost 100 fold differences in concentration, so 2 fold changes in concentration are likely irrelevant.
Also, IL-8 is only 1 of multiple CXCL that are chemotactic for neutrophils and none of these others were measured. A functional assay would account for these.
I am concerned that half the number of bacteria were added to the cultures when used in mixed population compared to monoinfections. I understand that the sum of the two different bacteria is the equivalent of the number of bacteria when used alone. But what was the titre after 4 hours? The monoinfections will possess greater numbers of bacteria than the same organism used in the mixed infection, in fact one of the mechanisms for the outcomes of the mixed infections may be the inhibition of bacterial growth.
No mechanism for the impact of one bacteria on another bacteria measured as stimulation of IL-8 secretion, was shown.
Tables 3 and 4 are unnecessary, the data can be reported in the text.
minor points:
IL-8 should be written as CXCL8
the manuscript needs to be edited for English
Some labelling on figure legends is in consistent or absent.
Reviewer 2 Report
The authors show an in vitro reduction of interleukin-8 response to Enterococcus faecalis by Escherichia coli in mixed infections. This topic is of interest.
Major comments
Experimental design is not appropriate:
- The E. coli strains were obtained from female patients suffering from cystitis and have been termed UPEC. The authors used MALDI-TOF MS for taxonomy and VITEK 2 for antibiograms. But they did not determine the real pathotypes of isolated E. coli. Are the strains really uropathogenic E. coli or are they urocolonizing E. coli? For example see: Eberly et al 2020 Feb 14;432(4):786-804. doi: 10.1016/j.jmb.2019.11.008. Epub 2019 Nov 30.
- For in vitro experiments, the concentrations of bacterial strains were diluted to 108 CFU/mL each. For co-infections authors used 10 microliter of each strain. For mono-infections authors used 20 microliter. Hence, they used 106 CFU/mL in co-infections and 2x106 CFU/mL in mono-infectios: the authors doubled the concentration in mono-infections. Why?
- Isolated E. coli were declared as "pathogenic" by the authors (= UPEC). Therefore, they used the non-pathogenic E. coli K12 as control. But authors did not use a non-pathogenic Enterococcus faecalis as control. Why? This reviewer recommend to use such an isolate, e.g. the non-pathogenic isolates Enterococcus faecalis Symbioflor 1, OB14 or OB15.
Minor comments
- Figures 1 and 4: Presentation of figures can be improved for a better view. E.g. use A), B), C), D).
- Check spelling.
- This reviewer want to see a few sentences about pathotypes of E. coli.

Round 2
Reviewer 1 Report
The authors did not grasp the depth of my concern over the histogram images showing CXCL8 concentrations. My understanding is that each bacteria may have been assayed for CXCL8 stimulation in separate experiments using different monolayers of bladder cells but in the figures results from different experiments are mixed together in comparisons. This is inappropriate. If all the assays were done in one large experiment with the same preparation of bladder cells at the same time there should only be one experiment and one figure. It has to be stated more clearly how these results were obtained. If they were done at different times on different bladder cell monolayers then what was done to try standardize the monolayer between experiments (e.g. passage number, cell density, viability check, same batch of bovine serum, etc)?
From MDPI | Research and Publication Ethics
"Image files must not be manipulated or adjusted in any way that could lead to misinterpretation of the information provided by the original image. Irregular manipulation includes 1) introduction, enhancement, moving, or removing features from the original image, 2) grouping of images that should obviously be presented separately (e.g., from different parts of the same gel, or from different gels)
Otherwise, there are still considerable deficiencies in the grammar/language. Here are some examples:
line 36: in both in- and -outpatients
line 84: and presence OF ≥ 108 CFU/mL
line 108: 2.3. Interleukine-8 Stimulation Assay
line 314: showed a different in vitro host-pathogen interaction with respect to the wildtype strains
Author Response
Dear Reviewer,
Please find the following answers to your observations:
- We apologize for not making it clear in our previous answer that the same monolayer was used in the stimulation experiments, all performed at the same time. For this reason there was no need to standardize cultures. We had to use two different packs of kits for CXCL8 measures, and showing these different detection curve, we preferred to stimulate the cells a second time with E. coli K12 strain. In fact, not only the supernatant from E.coli K-12 stimulation, but also that from uninfected cells were higher than the previous detection of CXCL8 in unifected and in E. coli K-12 infection supernatant. Even knowing that this could create perplexity we preferred, not being able to do it all over again, to represent the experiments as carried out and obtained. Obviously, we believe we have kept the meaning of the experiment intact by showing the results as ratios and percentages.
- Regarding the deficiencies in grammar / language we have double-checked all the text and made the necessary corrections
Reviewer 2 Report
I accept the modifications and explanations authors made.
Author Response
We are glad to have fulfilled the reviewer's requests